# Flopping for FLOPs: Leveraging Equivariance for Computational Efficiency

Georg Bökman [1]    David Nordström [1]    Fredrik Kahl [1]

## Abstract

Incorporating geometric invariance into neural networks enhances parameter efficiency but typically increases computational costs. This paper introduces new equivariant neural networks that preserve symmetry while maintaining a comparable number of floating-point operations (FLOPs) per parameter to standard non-equivariant networks. We focus on horizontal mirroring (flopping) invariance, common in many computer vision tasks. The main idea is to parametrize the feature spaces in terms of mirror-symmetric and mirror-antisymmetric features, i.e., irreps of the flopping group. This decomposes the linear layers to be block-diagonal, requiring half the number of FLOPs. Our approach reduces both FLOPs and wall-clock time, providing a practical solution for efficient, scalable symmetry-aware architectures.

## 1. Introduction

One of the main drivers of progress in deep learning is the scaling of compute. This idea is perhaps best summarized in Sutton's Bitter Lesson (Sutton, 2019). As Sutton states:

> Seeking an improvement that makes a difference in the shorter term, researchers seek to leverage their human knowledge of the domain, but the only thing that matters in the long run is the leveraging of computation.

It follows from the Bitter Lesson that a guiding principle for designing deep learning models should be to find simple models that scale well. In computer vision, this has proven fruitful and led to models such as the Vision Transformer (ViT) (Dosovitskiy et al., 2021). However, a straightforward critique of Sutton's argument is that leveraging domain knowledge can yield algorithms that better utilize computation. This paper explores common image symmetries that

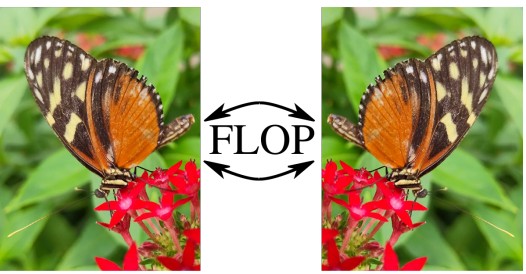

Figure 1: Common image classification tasks are invariant to flopping (horizontal mirroring). In our implementation, this invariance is enforced through equivariant network layers, halving the required floating-point operations (FLOPs).

enable more efficient computations. In a nutshell, our case is that equivariant neural networks can be simple models that scale well.

Many vision problems are invariant (or more generally equivariant) to horizontal mirroring, also known as *flopping*[1], and partially invariant to other geometric transformations such as rotations. For instance, both images in Figure 1 show a butterfly of the same species. Enforcing geometric invariance in a neural network architecture can be done by hard-coding the symmetry in each network layer, by enforcing specific weight sharing (Wood & Shawe-Taylor, 1996; Cohen & Welling, 2016). It has been shown in the literature that such architectural symmetry constraints typically improve the parameter efficiency of neural networks (Cohen & Welling, 2016; Bekkers et al., 2018; Weiler & Cesa, 2019). However, this comes at the cost of requiring more compute per parameter. For instance, Klee et al. (2023) found that rotation invariant ResNets typically outperformed ordinary ResNets on ImageNet-1K in the equal-parameter setting, but that the training time of the invariant ResNets far exceeded that of the ordinary ResNets. This is because the symmetry-induced weight sharing means that each trainable parameter is used in more computations.

In this paper, we show that invariant neural networks can achieve a comparable number of floating-point operations (FLOPs) per parameter to ordinary neural networks. Imple-

---

[1] Chalmers University of Technology. Correspondence to: Georg Bökman <bokman@chalmers.se>.

*Proceedings of the 42nd International Conference on Machine Learning*, Vancouver, Canada. PMLR 267, 2025. Copyright 2025 by the author(s).

[1] See Wikipedia article or Oxford definition.

menting these networks improves both FLOPs and actual wall-clock time. Our FLOP-efficient design is enabled by parametrizing features using irreducible representations of the flopping group.

Section 2 reviews related work on equivariant networks and Section 3 provides an exposition of flopping-equivariant networks for image classification, while Section A lays out the underlying mathematical background in detail. In Section 4, we design flopping-equivariant versions of popular modern vision architectures. Experiments in Section 6 on ImageNet-1K demonstrate that as network size scales up, flopping-equivariant models achieve comparable or improved classification accuracy for ResMLPs (Touvron et al., 2023), ConvNeXts (Liu et al., 2022) and ViTs (Dosovitskiy et al., 2021), while requiring only half the FLOPs.

## 2. Related Work

Our work is part of the broad field of geometric deep learning (Bronstein et al., 2021). In this section, we aim to discuss the most relevant work to the present paper, but we reference many other relevant works throughout.

### 2.1. Equivariant Networks for Image Input

Most work on equivariant networks on image input concerns convolutional networks (ConvNets) (Fukushima, 1975; Le-Cun et al., 1989). ConvNets are themselves equivariant to (cyclic) image translations. The ConvNets in this paper are built on ConvNeXt (Liu et al., 2022), which is a modern variant with state-of-the-art classification performance.

Making ConvNets equivariant to rotations and reflections was done by Cohen & Welling (2016) and Dieleman et al. (2016), with many follow-up works. The unifying framework of steerable $E(2)$-equivariant ConvNets (Cohen & Welling, 2017; Weiler & Cesa, 2019), subsumes much of the prior and subsequent work and proposes decomposing the feature spaces into irreps (irreducible representations, see Section A) of the symmetry group. Our ConvNeXt-variants are special cases of this general framework too.

Apart from ConvNets, there are works proposing attention- and transformer-based equivariant vision architectures (Romero et al., 2020; Xu et al., 2023). Most similar to our ViT-based networks are the $SO(2)$-steerable transformers by Kundu & Kondor (2024). The main differences are the group considered, the fact that they use complex-valued features, the nonlinearities used and the type of positional embeddings. However, the main idea of parametrizing the features in terms of irreps (following steerable ConvNets) and modifying the ViT architecture accordingly is the same in our ViTs and the one by Kundu & Kondor (2024).

### 2.2. Equivariant Networks for Other Input Than Images

Equivariant networks constitute a broad research direction. One early line of research is by Wood & Shawe-Taylor (1996) and going back earlier the topic connects to steerable filters (Knutsson & Granlund, 1983; Freeman & Adelson, 1991). Recent approaches include canonicalization (Kaba et al., 2023; Mondal et al., 2023), learned equivariance (Gupta et al., 2024), structured matrices (Samudre et al., 2025) et cetera. We will use the standard approach of enforcing each layer of the network to be equivariant through constraints on the weight matrices (Wood & Shawe-Taylor, 1996; Cohen & Welling, 2016).

There has been recent interest in the scaling properties of equivariant networks. Brehmer et al. (2024) find that the equivariant transformer GATr (Brehmer et al., 2023) outperforms non-equivariant transformers on a task of 3D rigid-body interactions, both in terms of parameter-efficiency and learning-efficiency, i.e., the number of FLOPs required during training.

Bekkers et al. (2024) propose a fast equivariant group convolutional network on point cloud input by using separable group convolutions on position orientation space $\mathbb{R}^3 \times S^2$ and parallelizing the message passing step. Their networks use ConvNeXt-style blocks, as do some of our networks. Follow-up work extends the message passing to use universal invariants (Bellaard et al., 2025), and experimentally analyzes the benefits of equivariance in different point cloud processing tasks (Vadgama et al., 2025).

In contrast to our networks, none of the mentioned works develop networks that are directly comparable with non-equivariant ones, with a similar FLOPs-per-parameter ratio.

## 3. Flopping Equivariance

We focus on horizontal mirroring invariance in image classification as a simple prototype task, aiming to inspire further research on scaling equivariant neural networks. Readers interested in a more abstract discussion in terms of general mathematical groups are referred to Section A. This section provides an accessible introduction to equivariant neural networks with minimal prerequisites. By emphasizing computational aspects, we aim to offer an intuitive motivation for our approach, making the key ideas easier to grasp.

The core idea is to split all feature maps in our network into two types: flopping-invariant features, which remain unchanged when the image is flopped, and flopping $(-1)$-equivariant features, which flip sign under flopping. By explicitly keeping track of how the features transform, it is possible to feed only invariant features into the final classification layer and guarantee flopping invariant predictions.

### 3.1. Linear Layers

Beyond ensuring flopping-invariant predictions, splitting features into invariant and $(-1)$-equivariant components also provides a significant computational advantage. Consider a linear mapping $W$ from $\mathbb{R}^c$ to $\mathbb{R}^d$. Assume that the first $c/2$ dimensions of the input are flopping invariant while the last $c/2$ dimensions are $(-1)$-equivariant, and likewise that the first $d/2$ dimensions of the output are invariant, while the last $d/2$ dimensions are $(-1)$-equivariant. We split the linear map $W$ into four parts,

$$\begin{pmatrix} y_1 \\ y_{-1} \end{pmatrix} = \begin{pmatrix} W_{1,1} & W_{1,-1} \\ W_{-1,1} & W_{-1,-1} \end{pmatrix} \begin{pmatrix} x_1 \\ x_{-1} \end{pmatrix} \tag{1}$$

where $x_1$ are the invariant input features and $x_{-1}$ the $(-1)$-equivariant input features and likewise for $y$. Now, by inspection, if any element of $W_{1,-1}$ is non-zero, then $y_1$ will change when $x_{-1}$ changes sign and if any element of $W_{-1,1}$ is non-zero, then $y_{-1}$ can not transform by multiplication with $-1$ jointly with $x_{-1}$. We conclude that $W_{1,-1}$ and $W_{-1,1}$ must be zero so that (1) actually reads

$$\begin{pmatrix} y_1 \\ y_{-1} \end{pmatrix} = \begin{pmatrix} W_{1,1} & 0 \\ 0 & W_{-1,-1} \end{pmatrix} \begin{pmatrix} x_1 \\ x_{-1} \end{pmatrix}. \tag{2}$$

This result reveals a key computational benefit: instead of performing a full $d \times c$ by $c$ matrix-vector multiplication, we only need two smaller ones of size $(d/2) \times (c/2)$ by $c/2$. This effectively **reduces the FLOPs by half** while preserving equivariance. More generally, the block-diagonalization from (1) to (2) follows from Schur's lemma (Lemma A.2) which applies to all finite symmetry groups. It is closely related to the fact that convolutions reduce into elementwise multiplications in the Fourier domain.

Linear layers of the form (2) are called *equivariant*. In general, any layer that preserves the transformation properties of the input is called equivariant. For instance, we can add a bias parameter to the equivariant linear layer if we enforce the bias to be zero for the $(-1)$-equivariant features. We give a more formal definition of equivariance in Section A, cf. (11). The stacking of equivariant layers to build a symmetry-respecting network is sometimes called the geometric deep learning blueprint (Bronstein et al., 2021), which goes back at least to the 90's with the works by Wood & Shawe-Taylor (1996). Using flopping invariant and $(-1)$-equivariant features is a special case of the steerable features proposed by Cohen & Welling (2017). However, general convolutional layers can not be be as straightforwardly block-diagonalized as in (2), since the convolutional kernels have a spatial extent. In particular, in the networks by Cohen & Welling (2017), the described computational boost is not realized, as the equivariant convolutions are implemented in terms of ordinary convolutions. We will discuss this in more depth in Section 4.4.

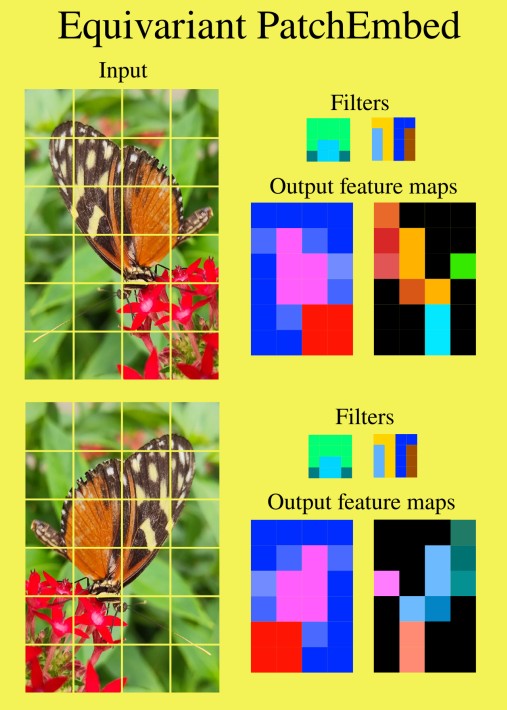

Figure 2: **Patch embedding layer.** The patch embedding (PatchEmbed) layer is common in modern networks, proposed with the ViT-model (Dosovitskiy et al., 2021). PatchEmbed is a convolutional layer with stride equal to the kernel size of the convolution filters. In our equivariant architectures, we enforce half of the filters to be symmetric and half to be antisymmetric. When the input image is flopped, the output feature map of a symmetric filter is flopped as well. The output feature map of an antisymmetric filter is flopped and changes sign. Half of the features output from our PatchEmbed layer are flopping invariant, while half are $(-1)$-equivariant, enabling efficient equivariant processing in subsequent linear layers, as detailed in Section 3.1.

Three important questions remain and will be discussed next. First, how do we obtain features that are invariant and $(-1)$-equivariant? Second, how do we design the other parts of the network? Third, do we reduce representation power by forcing our features to be invariant and $(-1)$-equivariant?

### 3.2. Patch Embedding Layer

Obtaining invariant and $(-1)$-equivariant features from an image is easy. One straightforward manner is to convolve the image with symmetric and antisymmetric filters as illustrated in Figure 2. Such convolutions can be incorporated as the first layer in convolutional neural networks and vision transformers alike (the so-called patch embedding layer, or "PatchEmbed") and correspond to the lifting layer in an equivariant ConvNet (Cohen & Welling, 2016).

### 3.3. Non-Linear Layers

Non-linear layers of the neural network also need to be equivariant, i.e. respect the transformation rules of the features. For instance, common layers in modern networks include layer normalization (Ba et al., 2016), activation functions such as GELU (Hendrycks & Gimpel, 2016), as well as multi-head self-attention (Bahdanau et al., 2015; Vaswani et al., 2017). Layer normalization is equivariant without modification since the norm of neither $x_1$ nor $x_{-1}$ changes as the input is flopped. For pointwise activation functions $\sigma$, we can create equivariant versions by computing the output $y_1, y_{-1}$ through

$$
\begin{aligned}
s &\hookleftarrow \sigma\big((x_1 + x_{-1})/\sqrt{2}\big), \quad t \hookleftarrow \sigma\big((x_1 - x_{-1})/\sqrt{2}\big) \\
y_1 &\hookleftarrow (s+t)/\sqrt{2}, \quad y_{-1} \hookleftarrow (s-t)/\sqrt{2}.
\end{aligned}
\tag{3}
$$

The reader is encouraged to check that when $x_{-1}$ is multiplied by $-1$, $y_1$ remains constant and $y_{-1}$ is multiplied by $-1$. Here (3) can be interpreted as transforming from the "Fourier" domain to the "spatial" domain ($s$ and $t$ permute when $x_{-1}$ changes sign), applying $\sigma$ there and transforming back. We include the factor $1/\sqrt{2}$ to preserve the norm of the features. The computation in (3) is heavier than just applying $\sigma$, but this overhead is negligible compared to, for instance, linear layers.

### 3.4. Attention

For self-attention, if the queries $q_i$ and keys $k_j$ are split into flopping invariant and $(-1)$-equivariant features, their dot products form an invariant quantity:

$$
a_{ij} = q_{i,1} \cdot k_{j,1} + q_{i,-1} \cdot k_{j,-1} = q_{i,1} \cdot k_{j,1} + (-q_{i,-1} \cdot -k_{i,-1}).
\tag{4}
$$

The $a_{ij}$'s are then normalized using softmax as in standard scaled dot-product attention and the same attention score is multiplied by both invariant and $(-1)$-equivariant values $v_{j,\pm 1}$ to obtain the output.

### 3.5. Limitations of Equivariant Networks

Finally, we need to discuss the limitations put on the model by enforcing each layer to be equivariant. Prior work has shown experimentally, and theoretically in some special cases, that ordinary networks learn to be approximately equivariant by training on symmetric data (Lenc & Vedaldi, 2015; Olah et al., 2020; Gruver et al., 2023; Bruintjes et al., 2023; Bökman & Kahl, 2023; Marchetti et al., 2024)[2]. What is meant by this is that for a given output feature space of

a particular layer in the network, there exists a "steering matrix" $A$, such that when the input is flopped, the output feature is (approximately) transformed by $A$. Since flopping an image twice returns the original image, we must have that $A^2 = I$, which means that $A$ can be diagonalized as $A = QDQ^{-1}$ with $D$ diagonal containing only entries of $\pm 1$. (The generalization of this diagonalization is called the isotypical decomposition and is possible for more general symmetry groups as well, due to Maschke's theorem (Theorem A.1).) In other words, by changing the basis that we parametrize the network features in by $Q$, we get features that are invariant and $(-1)$-equivariant as before. Therefore, restricting the network to be equivariant is not really limiting what features are learnt if the equivariance is learned even without this restriction.

The emergence of equivariance from data has sometimes been given as an argument against hard-coding equivariance in the network. We want to turn this argument around (or flop it) by saying that if the network learns equivariance in any case, we might as well hard-code it and take advantage of the computational benefits that come for free. It should be mentioned, however, that it is not obvious that the choice with an equal number of invariant and $(-1)$-equivariant features is optimal. In fact, Bökman & Kahl (2023) found that networks often learn more invariant features than equivariant features for classification tasks, while Bökman et al. (2024) found that for keypoint description, networks often learn an equal amount of invariant and equivariant features[3].

It is also not obvious that an equivariant layer has as good a parametrization as an ordinary layer for network training, particularly when using standard training recipes developed for ordinary layers. Recent work by Pertigkiozoglou et al. (2024) shows that it may be possible to improve the training of equivariant networks by relaxing the equivariance constraint during training. In any case, the main aim of the present paper is to demonstrate that equivariant networks, when parametrized right, are as FLOP-efficient per parameter as ordinary networks. We leave the investigation of optimal design and training of equivariant networks as orthogonal research directions for future work.

### 3.6. Generalization to Other Groups

The argument given for computational savings in linear layers can be generalized to other groups than the flopping group. In Section A, we go through the mathematical background in detail. Importantly, whenever the representations

---

[2]Some prior work also found evidence in the other direction, i.e., that weight symmetries do not always appear in networks trained on symmetric data (Moskalev et al., 2023). Thus, it is a partially unsettled question to what extent and in which situations equivariance can be learned from data. As we argue in this section, it is however the case that even if we take the least favourable view

for motivating hard-coding equivariance—saying that it can be learned from data—there is still a clear computational argument in favour of hard-coding.

[3]Of course, nothing guarantees that training an ordinary network with gradient-based optimization yields the optimal allocation of invariant and equivariant features.

acting on the feature spaces are completely reducible (i.e. a direct sum of irreps), linear layers can always be block-diagonalized by parameterizing them in terms of the irreps. The computational savings depend on (a) the number of irreps, (b) the dimensionality of the irreps, and (c) whether the real irreps are of real, complex or quaternion type. Computational savings are possible in general but the number of FLOPs per parameter will be higher for irreps with dimension larger than 1.

## 4. Modern Neural Networks for Image Data

Since the development of Highway nets (Srivastava et al., 2015) and ResNets (He et al., 2016a), modern neural networks for image data consist of stacking residual blocks

$$B(x) = x + \phi(x). \tag{5}$$

Often, $\phi$ is a composition of a normalization layer, such as batch normalization or layer normalization, and a computational sequence, such as a two-layer MLP, a self-attention layer or a two-layer ConvNet. The main reason for using residual blocks is to facilitate network training by enabling gradient propagation through many layers.

We cover three architectures based on (5), and outline the minimal changes to make them flopping-equivariant. We start with the simplest architecture, ResMLP (Touvron et al., 2023), working our way through ViT (Dosovitskiy et al., 2021) and finally ConvNeXt (Liu et al., 2022).

Figure 3 shows a schematic of the network design. All three network families incorporate a patch embedding layer ("PatchEmbed") as the first layer, as outlined in Section 3.2. We replace all linear layers (except the depthwise convolutions in ConvNeXt) by block diagonal linear layers as in (2) and use equivariant layer norm, attention and GELU-nonlinearities as discussed in Section 3. Further special layers will be discussed in the subsections below.

We chose architectures and training recipes based on three criteria: the architectures should be well established, a PyTorch implementation should be available, and training recipes for ImageNet-1K should be available, as larger datasets are out of reach for our computational budget.

### 4.1. ResMLP

Tolstikhin et al. (2021), Melas-Kyriazi (2021) and Touvron et al. (2023) independently proposed a simplification of vision transformers, where the attention layers are replaced by MLPs over the patch dimension. These model types are most often denoted as MLP-Mixers in the literature, but as we use the ResMLP version and training recipe by Touvron et al. (2023), we will usually refer to the models as ResMLP.

To make the linear layers over the patch dimension equivari-

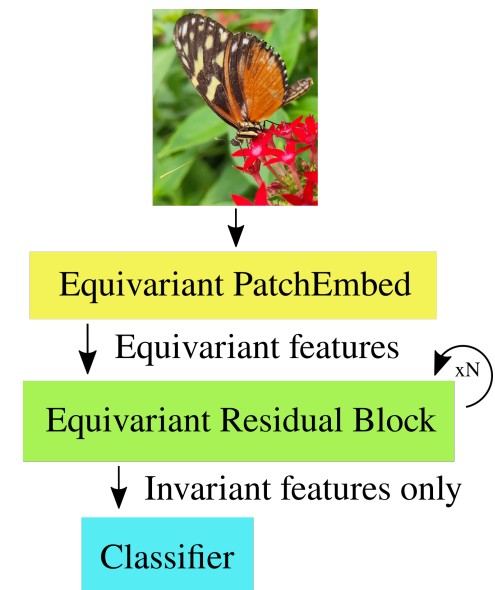

Figure 3: **Schematic.** Our architectures follow the basic geometric deep learning blueprint (Wood & Shawe-Taylor, 1996; Cohen & Welling, 2016; Bronstein et al., 2021) in combination with modern vision models (Dosovitskiy et al., 2021; Liu et al., 2022; Touvron et al., 2023). The equivariant PatchEmbed is described in Section 3.2. The residual blocks differ per architecture and are explained in Section 4. The classifier head is a single linear layer.

ant and efficient, we decompose the features not only into invariant and equivariant in the channel dimension, but also in the patch dimension. This is detailed in Section C.

One unique aspect of ResMLP is that it doesn't normalize features, but instead uses an affine layer (over the channel dimension), in symbols

$$\text{Aff}_{\alpha,\beta}(x) = \text{Diag}(\alpha)x + \beta, \tag{6}$$

where $\alpha$ and $\beta$ are learnable weight vectors. To make (6) flopping-equivariant we only need to set $\beta$ to constant 0 for the $(-1)$-equivariant features.

Touvron et al. (2021) proposed *LayerScale*, a learnable diagonal matrix applied on the output of each residual block, initialized close to 0. This is just an affine layer as in (6) without bias $\beta$ and is also included in ResMLP.

The ResMLP-blocks are of the form $B = B_2 \circ B_1$ with

$$\begin{aligned} B_1(x) &= x + \text{LS}(\text{FC}_{\text{patches}}(\text{Aff}(x))), \\ B_2(x) &= x + \text{LS}(\text{MLP}_{\text{channels}}(\text{Aff}(x))), \end{aligned} \tag{7}$$

where LS denotes layer scale, FC a fully connected (block-diagonal in our case) layer and MLP two fully connected (block-diagonal) layers separated by GELU.

## 4.2. ViT

Vision transformers (ViTs) use transformer encoder blocks (Vaswani et al., 2017) as residual blocks. The layer closely follows the original transformer implementation and only deviates by the layer normalization being applied before the computation, as proposed by He et al. (2016b).

In particular, the residual block can be expressed as $B = B_2 \circ B_1$ with

$$
\begin{aligned}
B_1(x) &= x + \text{LS}(\text{MSA}(\text{LN}(x))), \\
B_2(x) &= x + \text{LS}(\text{MLP}(\text{LN}(x))),
\end{aligned}
\tag{8}
$$

where LS is LayerScale, LN layer normalization, MSA is multi-head self attention and MLP consists of two fully connected (block diagonal in our case) layers separated by GELU. LayerScale was not part of the original ViTs but is included in the ViTs used as a benchmark in this paper as we follow the training recipe by Touvron et al. (2022).

To make ViT flopping-equivariant we further modify the positional encoding to consist of half channels that are symmetric about the middle of the image and half channels that are anti-symmetric, thus forming invariant and $(-1)$-equivariant features. Positional encodings are added to the features after the initial PatchEmbed layer. We also enforce the classification token appended to the embedded image patches in ViTs to be constant zero for the $(-1)$-equivariant features. Multi-head self attention is made equivariant by using block-diagonal linear layers for the projections to queries, keys and values followed by attention as described in Section 3.4.

To experiment with varying the group representations in the intermediate layers, we implement ViT-versions that use only invariant features in half the layers. After the first half of the network, we map the $d/2$ invariant dimensions linearly to $d$ invariant dimensions and throw away the $(-1)$-equivariant features. The subsequent layers are then ordinary ViT-layers, which preserve invariance. In these last layers, we do not get any computational saving compared to the baseline since the block-diagonalization in (2) will not have any $W_{-1,-1}$ or 0-blocks. We denote these half-way invariantized networks by $\mathcal{I}(\text{ViT})$.

Following Weiler & Cesa (2019), we also implement a hybrid model[4] where the first half of the residual blocks are equivariant and the second half are standard blocks. The intuition is that inductive bias is most useful in early network layers for learning general features in all orientations, while the last layers use more specialized processing. As this is not the focus of the paper, we include only hybrid versions of the ViTs, denoted $\mathcal{H}(\text{ViT})$, to test the limits of scaling equivariant layers.

---

[4]These are called "restricted" by Weiler & Cesa (2019).

## 4.3. ConvNeXt

ConvNeXt, proposed by Liu et al. (2022), builds on the ideas of the ViT architecture, outlined above, to create a modern family of pure ConvNet models that compare favorably with ViTs in terms of accuracy and scalability. The isotropic ConvNeXt architecture, denoted Convnext (*iso.*), even more closely resembles ViT as it has no downsampling layers and keeps the same number of patches/pixels at all depths. We use the isotropic ConvNeXt architecture in this paper as it allows for a more efficient implementation. The reason is that we do not have efficient implementations of symmetric and antisymmetric convolutions for the downsampling layers; refer to Section 4.4.

The main residual block is composed of a depthwise convolution, using a $7 \times 7$ convolutional kernel, followed by two $1 \times 1$-convolutions, separated by a GELU non-linearity. The residual block can be expressed as $B = B_2 \circ B_1$ with

$$
\begin{aligned}
B_1(x) &= x + \text{LN}(\text{DwConv7x7}(x)), \\
B_2(x) &= x + \text{LS}(\text{Conv1x1}(\text{GELU}(\text{Conv1x1}(x)))),
\end{aligned}
\tag{9}
$$

where Dw stands for depthwise.

Out of the depthwise convolutions, half are set to be symmetric and half to be antisymmetric. They are alternated so that out of the invariant input features, half are convolved with symmetric filters and half with antisymmetric—generating invariant and $(-1)$-equivariant features respectively, and vice versa for the $(-1)$-equivariant inputs. The $1 \times 1$ convolutions are implemented using the block diagonal form (2).

## 4.4. On Efficient Steerable Convolutions

In prior work (Weiler & Cesa, 2019), steerable convolutions were implemented by inputting equivariant filters in ordinary convolution software routines. This did not provide any computational benefit over non-equivariant filters, which use the same routines. Our implementations are more efficient because the ConvNeXt architecture utilizes depthwise convolutions followed by $1 \times 1$-convolutions, and the $1 \times 1$-convolutions are pixel-wise linear layers that can be decomposed into block-diagonals as in (2). There are previous works that use separable convolutions in equivariant networks, but they parametrize the features in the spatial domain rather than in terms of irreps and do not obtain a computational advantage over ordinary separable convolutions (Lengyel & van Gemert, 2021; Knigge et al., 2022).

It would actually be possible to implement more efficient steerable $k \times k$-convolutions as well. For illustration, consider a one-dimensional correlation of a symmetric filter $[a, a]$ by a signal $[x, y, z]$. We can reduce the number of

required FLOPs by writing the output as

$$[a, a] \star [x, y, z] = [ax + ay, ay + az]$$
$$= [a(x + y), a(y + z)]. \qquad (10)$$

Even more efficiently, one can use so-called Winograd schemes specific for symmetric or anti-symmetric filters (Winograd, 1980). We do not pursue this optimization in the present paper, as it requires significant effort into writing GPU-code, but we believe that it is a promising direction for further improving the runtime of steerable ConvNets. Thus, the FLOP and throughput values reported for $\mathcal{E}(\text{ConvNeXt } (iso.))$ in Table 1 do not include the possible further reduction obtainable from more efficient symmetric and anti-symmetric depthwise convolutions.

## 5. Equivariant vs. Non-Equivariant Networks

There are a couple of different ways that equivariant networks can be compared to ordinary networks. Cohen & Welling (2016) and most later works compare an equivariant network with $M$ trainable parameters with ordinary networks with $M$ parameters. This is a fair comparison from a learning theory perspective since the parameters can then store the same amount of information. Subsequent works noted that this is unfair from a computational perspective, (Weiler & Cesa, 2019; Klee et al., 2023; Roos & Kroon, 2024), because of the fact that the equivariant ConvNets had the same computational cost as ordinary network with many more parameters, since as mentioned they used the same convolution routines.

The first measure for computational cost that we will use in this paper is the number of FLOPs required for a forward pass of the network. This represents a bound on how efficient the network can be made. In the limit, as feature spaces grow and dense linear layers dominate the compute, the number of FLOPs accurately predicts how fast the network will run on modern GPUs. FLOPs are therefore used for analyzing the compute scaling of large networks (Kaplan et al., 2020). However, the most direct measure of computational cost is the network's throughput in terms of images per second. This is a somewhat inconsistent measure as it depends heavily on the hardware used and can sometimes vary drastically with minor implementation changes. We believe that the best way to compare networks is to report all three measures: number of parameters, FLOPs and throughput, as is usually done in the literature on non-equivariant networks. One of the main points of this paper is to illustrate that contrary to popular belief, flopping equivariant networks can be designed to have approximately the same number of FLOPs per parameter as ordinary networks (see Figure 4).

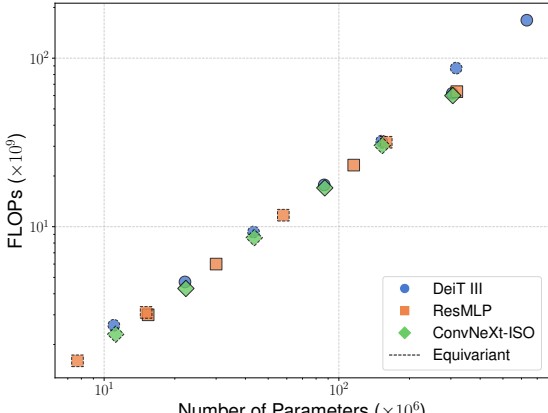

Figure 4: **FLOPs scaling by model size.** Comparing the number of FLOPs to the number of parameters in the model.

## 6. Experiments

In this section, we evaluate the effectiveness of flopping-equivariant networks. We first briefly discuss the setting of the experiments. Then we compare the efficiency and accuracy of equivariant versions of ResMLPs, ViTs and ConvNeXts from Section 4 to their non-equivariant counterparts. For a given architecture X, the equivariant version is $\mathcal{E}(\text{X})$. $\mathcal{E}(\text{X})$ always has around half the number of trainable parameters and FLOPs of X due to the block-diagonalization of the weight matrices (2). We will release code and weights at github.com/georg-bn/flopping-for-flops.

**Dataset.** We benchmark our model implementations on the ImageNet-1K dataset (Deng et al., 2009; Russakovsky et al., 2015; Recht et al., 2019), which includes 1.2M images evenly spread over 1,000 object categories.

**Hyperparameters.** We use the same training recipes as the baselines. The complete set of hyperparameters can be found in Table 2 in the appendix.

**Implementation.** The major building blocks of modern deep learning, including linear layers, convolution layers and attention layers, are all implemented using general matrix multiplications (GEMMs). Modern GPUs enable efficient GEMMs, typically implemented in a tiled fashion, processing blocks of the output matrix in parallel. Consequently, reducing the linear layers into block diagonals does in principle not hinder GPU utilization, as the same tile-GEMMs can be used for the full product (1) or the block-diagonal (2) (with the latter requiring half as many). We use the PyTorch (Paszke et al., 2019) compiler to obtain efficiently running networks, but writing custom GPU-code would likely further improve the throughput.

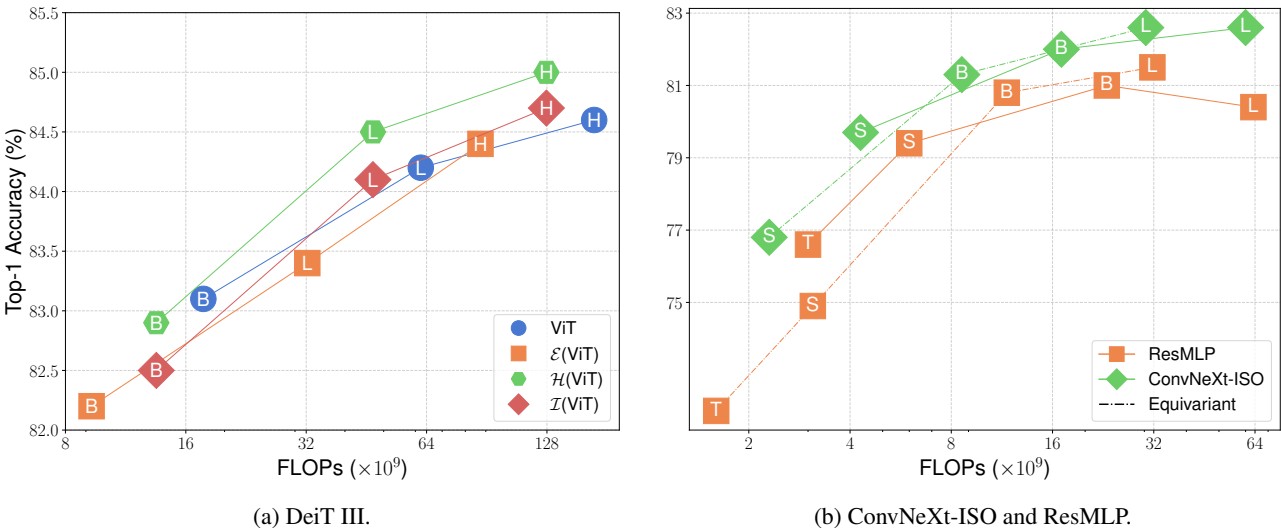

(a) DeiT III.        (b) ConvNeXt-ISO and ResMLP.

Figure 5: Validation accuracy on ImageNet-1K versus model complexity as measured by the number of FLOPs required per image for a forward pass. Letters indicate the respective model sizes, see Tables 1 and 3 for details.

### 6.1. Results and Discussion

Figure 4 shows that our networks maintain a comparable number of floating-point operations (FLOPs) per parameter to standard networks.

In prior work, image classification is often contextualized as a trade-off between accuracy and measures of model complexity, such as FLOPs and number of parameters. To that end, in Table 1, we highlight both the theoretical complexity, i.e. model size, and FLOPs, as well as the empirical speed as measured by the throughput (images processed per second), and the Top-1 classification accuracy on ImageNet-1K. Our results demonstrate that as we increase the scale of the networks, flopping-equivariant networks achieve competitive classification accuracy while requiring half the FLOPs, as highlighted in Figure 5, and showcase higher throughput compared to baseline implementations. The hybrid model $\mathcal{H}(\text{ViT-H})$ achieves the best accuracy of all models, aligning with the results for hybrid models by Weiler & Cesa (2019). One reason the smaller equivariant models underperform in terms of accuracy is that they are quite limited in the number of parameters—the regularization methods in the training recipes for the corresponding non-equivariant models (with more parameters) may be too heavy. We leave the compute-heavy task of optimizing the training recipes for future work, as we believe that our experiments clearly demonstrate the main point of this paper—that equivariant networks can be scalable.

As shown in Figure 6, the relative throughput improvement of the equivariant networks becomes more pronounced as the size of the model increases, which can be attributed to a higher proportion of time spent on the linear layers, with the

computational overhead from less optimized kernels becoming less significant. This is best illustrated by the superior speed of the equivariant ResMLP models for which linear layers dominate the computation. Our networks have no throughput improvement for attention or depthwise convolutions, which, despite not contributing the most FLOPs, are sometimes comparable in runtime to the dense layers. Future more efficient implementations of these layers, and further scaling of the embedding dimension, would pronounce the gain in throughput for the equivariant models.[5]

Notably, for small models, such as ViT-S and ConvNeXt-S, the equivariant versions have a worse throughput. For making these efficient, custom GPU-code is likely required.

### 6.2. Experimental Limitations

We did not do any hyperparameter tuning (so the baselines we compare against are at an advantage). Particularly, we noted that some of the training runs were quite unstable: We were not able to get $\mathcal{E}(\text{ViT-S})$ to converge despite trying different random seeds, and the result of the non-equivariant ResMLP-L24 (the only model size that we introduced ourselves and thus trained the baseline for) was unexpectedly low. We only ran one training per model, as we had a limited compute budget. This practice aligns with the prior work. We limited the experiments to supervised training on ImageNet-1K, while the best results are usually obtained with pretraining on Imagenet-21K or distillation of larger

---

[5]The reader will notice a flattening out of the rate of improvement between ViT-L and -H. ViT-H uses patch size $14 \times 14$ while the other ViTs use patch size $16 \times 16$. Smaller patch sizes means more patches and more compute in the attention layers.

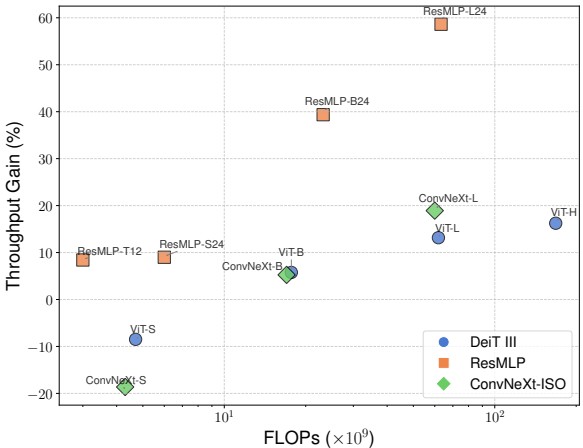

Figure 6: **Throughput gain based on model size.** The throughput gain as measured by the percentage difference between the equivariant and baseline implementations, placed according to the baseline's number of FLOPs.

models. The hardware and software used for our training runs differed from the baselines, see Section B.

## 7. Conclusion

In this paper, we have introduced flopping-equivariant neural networks that maintain a comparable number of FLOPs per parameter to standard non-equivariant networks. We showed that these have increased computational efficiency not only in terms of FLOPs, but also actual throughput.

The considered task of upright image classification is in a sense the least interesting for equivariant networks as the symmetry group is small and the output is invariant. We believe that future efforts should be directed towards equivariant networks as general visual feature extractors. Classic and recent works have already demonstrated the utility of having equivariant features for downstream vision tasks (Lowe, 2004; Loy & Eklundh, 2006; Lee et al., 2023; Bökman et al., 2024; Garrido et al., 2024).

We hope that efficient equivariant networks can be adopted into modern vision backbones and that this work prompts more researchers to view equivariant architectures as not only theoretically compelling but also practically useful, not only for parameter efficiency in the small data regime but also for compute efficiency in the large data regime.

Table 1: **Classification with Imagenet1k training.** We contrast our flopping-equivariant networks to the originals using the same training recipes. The equivariant version of architecture X is denoted $\mathcal{E}(X)$. $\mathcal{H}(X)$ is a hybrid model with equivariant layers for the first half of the network. The baselines are not rerun, instead we show the results reported in their respective papers. The throughput and peak memory are measured on a single A100-40GB GPU with batch size fixed to 64 and compiled networks running mixed precision forward passes with no gradients.

| Architecture | params $(\times 10^6)$ | throughput (im/s) | FLOPs/img $(\times 10^9)$ | Peak Mem (MB) | Top-1 Acc. |
|---|---|---|---|---|---|
| **MLP-Mixers (ResMLP (Touvron et al., 2023))** | | | | | |
| ResMLP-L24[1] | 318.1 | 1107 | 63.3 | 1778 | 80.4 |
| $\mathcal{E}$(ResMLP-L24) | 159.2 | 1756 | 31.7 | 1056 | **81.5** |
| ResMLP-B24 | 115.7 | 2482 | 23.2 | 779 | 81.0 |
| $\mathcal{E}$(ResMLP-B24) | 58.0 | 3459 | 11.7 | 519 | 80.8 |
| ResMLP-S24 | 30.0 | 6445 | 6.0 | 320 | 79.4 |
| $\mathcal{E}$(ResMLP-S24) | 15.1 | 7025 | 3.1 | 249 | 74.9 |
| ResMLP-T12 | 15.4 | 12133 | 3.0 | 264 | 76.6 |
| $\mathcal{E}$(ResMLP-T12) | 7.7 | 13154 | 1.6 | 221 | 72.0 |
| **Vision Transformers (DeiT III (Touvron et al., 2022))** | | | | | |
| ViT-H | 632.1 | 431 | 168.0 | 3366 | 84.6 |
| $\mathcal{H}$(ViT-H) | 474.2 | 466 | 127.6 | 3231 | **85.0** |
| $\mathcal{I}$(ViT-H) | 474.2 | 462 | 127.6 | 3231 | 84.7 |
| $\mathcal{E}$(ViT-H) | 316.1 | 501 | 87.3 | 2598 | 84.4 |
| ViT-L | 304.4 | 1064 | 61.9 | 1726 | 84.2 |
| $\mathcal{H}$(ViT-L) | 228.3 | 1123 | 47.0 | 1743 | 84.5 |
| $\mathcal{I}$(ViT-L) | 228.3 | 1123 | 47.0 | 1743 | 84.1 |
| $\mathcal{E}$(ViT-L) | 152.2 | 1204 | 32.2 | 1459 | 83.4 |
| ViT-B | 86.6 | 3088 | 17.7 | 777 | 83.1 |
| $\mathcal{H}$(ViT-B) | 65.0 | 3162 | 13.5 | 936 | 82.9 |
| $\mathcal{I}$(ViT-B) | 65.0 | 3163 | 13.5 | 921 | 82.5 |
| $\mathcal{E}$(ViT-B) | 43.3 | 3266 | 9.3 | 828 | 82.2 |
| ViT-S | 22.1 | 7174 | 4.7 | 338 | 80.4 |
| $\mathcal{E}$(ViT-S) | 11.0 | 6565 | 2.6 | 417 | † |
| **Convolutional Networks (ConvNeXt (Liu et al., 2022))** | | | | | |
| ConvNeXt-L (*iso.*) | 305.9 | 1284 | 60.0 | 1420 | **82.6** |
| $\mathcal{E}$(ConvNeXt-L (*iso.*)) | 153.1 | 1527 | 30.3 | 935 | **82.6** |
| ConvNeXt-B (*iso.*) | 87.1 | 3890 | 17.0 | 540 | 82.0 |
| $\mathcal{E}$(ConvNeXt-B (*iso.*)) | 43.6 | 4094 | 8.6 | 450 | 81.3 |
| ConvNeXt-S (*iso.*) | 22.3 | 9649 | 4.3 | 226 | 79.7 |
| $\mathcal{E}$(ConvNeXt-S (*iso.*)) | 11.2 | 7851 | 2.3 | 220 | 76.8 |

[1]ResMLP-L24 was trained by us as a baseline.
†Failed to converge.

## Impact Statement

This paper presents work whose goal is to advance the field of Machine Learning. There are many potential societal consequences of our work, none which we feel must be specifically highlighted here.

## Acknowledgements

This work was supported by the Wallenberg Artificial Intelligence, Autonomous Systems and Software Program (WASP), funded by the Knut and Alice Wallenberg Foundation. The computational resources were provided by the National Academic Infrastructure for Supercomputing in Sweden (NAISS) at C3SE, partially funded by the Swedish Research Council through grant agreement no. 2022-06725, and by the Berzelius resource, provided by the Knut and Alice Wallenberg Foundation at the National Supercomputer Centre.

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

# A. Groups, Representations and Schur's Lemma

This section is not required reading to understand the main takeaway of the paper. It serves to describe how the content of Section 3 can be formalized and generalized using group theory. The theory content of this section is common knowledge in geometric deep learning (Bronstein et al., 2021; Gerken et al., 2023; Weiler et al., 2023) and part of standard representation theory courses (Serre, 1977). Our contribution is to highlight the computational aspect more than has perhaps been done in previous work.

We use the language of group representations to facilitate a precise discussion of symmetries. Groups are the mathematical abstraction of sets of symmetry transformations, such as the flopping of images. A *group* $(G, \circ)$ is a set $G$ together with an associative binary composition operation $\circ$. To be a group it has to have (i) a unit element $u \in G$ such that $u \circ g = g \circ u = g$ for all $g \in G$ and (ii) for each $g \in G$ an inverse $g^{-1} \in G$ such that $g \circ g^{-1} = g^{-1} \circ g = u$. For image flopping, the mathematical group is called $D_2$ (the dihedral group with two elements[6]) and consists of the identity $u$ and another element $h$, which are both their own inverse (flopping an image twice returns the original image).

A *group representation* concretises the group as a set of invertible matrices. We denote the set of invertible real $n \times n$ matrices by $\mathrm{GL}(n)$, and a representation of $G$ is a map $\rho : G \to \mathrm{GL}(n)$, that encodes the composition operation as matrix multiplication: $\rho(g_1 \circ g_2) = \rho(g_1)\rho(g_2)$. It follows from $\rho(u) = \rho(u \circ u) = \rho(u)\rho(u)$ that $\rho(u) = I$ is the identity matrix. Group representations enable us to discuss the same group acting on vector spaces of different dimensions. Therefore, group representations are useful to describe what happens to internal neural network features when the input is transformed. Specifically, a function $f$ is said to be *equivariant* with respect to representations $\rho_1$ and $\rho_2$ if

$$f(\rho_1(g)x) = \rho_2(g)f(x). \tag{11}$$

If $f$ consists of the first couple of layers of an equivariant neural network, and $\rho_1(h)$ is the permutation of pixels that flops an image $x$, then (11) says exactly what happens to the internal features of the neural network as the input image is flopped—the features are transformed by $\rho_2(h)$.

Following the geometric deep learning blueprint (Bronstein et al., 2021), to design flopping-invariant neural networks $f = f_N \circ \cdots \circ f_2 \circ f_1$, we enforce each network layer $f_i$ to be flopping-equivariant. The last layer is designed be flopping invariant, which is the special case of (11) where $\rho_2(g) \equiv I$. Designing an invariant neural network hence involves a choice of intermediate group representation $\rho_i$ for the output of each layer. We can write down all possible

choices of $\rho_i$ in terms of fundamental building blocks called irreducible representations.

An irreducible representation (*irrep*) $\varrho : G \to \mathrm{GL}(n)$ is a representation of $G$ such that the only subspaces of $\mathbb{R}^n$ that are invariant under all $\varrho(g)$ are $\mathbb{R}^n$ itself and the 0-vector space[7]. We will use the notation $\rho$ for representations in general and $\varrho$ for irreducible representations.

Representations that only differ by a change of basis $Q$ are not qualitatively different mathematically and are called *isomorphic*. Maschke's theorem implies that the feature spaces in equivariant neural networks are always isomorphic to concatenations of feature spaces that irreps act on.

**Theorem A.1** (Maschke's theorem). *Every representation $\rho : G \to \mathrm{GL}(n)$ of a finite group $G$ is a direct sum of irreps.*

Maschke's theorem means that for every representation $\rho$, there exists a change of basis matrix $Q \in \mathrm{GL}(n)$, non-isomorphic irreps $\varrho_i$ ($i = 1, \ldots, m$) and multiplicities $k_i \in \mathbb{Z}_{\geq 0}$ such that

$$\rho(g) = Q \left( \bigoplus_{i=1}^{m} \varrho_i(g)^{\oplus k_i} \right) Q^{-1}, \tag{12}$$

where $\bigoplus$ denotes the direct sum, i.e. assembling the matrices $\varrho_i(g)^{\oplus k_i}$ into a block-diagonal matrix and $X^{\oplus k}$ means a block-diagonal matrix with $k$ copies of $X$ along the diagonal. (12) is called the isotypical decomposition of $\rho$.

Having characterized all representations in terms of irreps, we next characterize all equivariant linear maps through *Schur's lemma*. Given two real representations $\rho_1 : G \to \mathrm{GL}(n_1)$ and $\rho_2 : G \to \mathrm{GL}(n_2)$, we denote the set of linear maps $A : \mathbb{R}^{n_1} \to \mathbb{R}^{n_2}$ that are equivariant under $\rho_1$ and $\rho_2$ by $\mathrm{Hom}(\rho_1, \rho_2)$. $\mathrm{Hom}(\rho_1, \rho_2)$ is a vector space since we can add and multiply by scalars without changing the equivariance property.

**Lemma A.2** (Schur's lemma). *Let $\varrho_1$ and $\varrho_2$ be two real irreps of a group $G$.*

1. *If $\varrho_1$ and $\varrho_2$ are not isomorphic, then $\mathrm{Hom}(\varrho_1, \varrho_2)$ contains only the zero-map.*

2. *If $\varrho_1$ and $\varrho_2$ are isomorphic, then $\mathrm{Hom}(\varrho_1, \varrho_2)$ is either 1-, 2- or 4-dimensional.*

The dimension in case 2 depends on whether the irrep $\varrho_1$ is of so-called real, complex or quaternary type (Serre, 1977).

We can put Maschke's theorem and Schur's lemma to work in a very useful standard corollary.

---

[6]Other names for $D_2$ are $C_2$, $\mathbb{Z}_2$, $S_2$ and (confusingly) $D_1$.

[7]This definition can also be stated for complex vector spaces and infinite vector spaces, but we will be satisfied with finite-dimensional real vector spaces in this paper.

**Corollary A.3.** *If $\rho_1 : G \to \mathrm{GL}(n_1)$ and $\rho_2 : G \to \mathrm{GL}(n_2)$ are real representations of a finite group $G$, then there are $Q_1 \in \mathrm{GL}(n_1)$ and $Q_2 \in \mathrm{GL}(n_2)$ such that for any $A \in \mathrm{Hom}(\rho_1, \rho_2)$, $Q_2^{-1} A Q_1$ is block-diagonal with blocks only mapping between isomorphic irreps.*

For discussing image flopping, it is useful to know that $D_2$ has only two irreps, both one-dimensional, defined by $\varrho_1(h) := 1$ (the "trivial" irrep) and $\varrho_{-1}(h) := -1$ (the "sign-flip" irrep). Therefore, by Corollary A.3, every equivariant linear map decomposes into a block-diagonal map with two blocks. Working in this block-diagonal basis hence gives a computational advantage. Features transforming by $\varrho_1$ are those called invariant and those transforming by $\varrho_{-1}$ are called $(-1)$-equivariant in Section 3.

For larger groups than $D_2$, the number of irreps are more numerous and so the computational savings in linear layers can be even larger. However, for groups with irreps of dimension $> 1$ the FLOPs-per-parameter will be worse and with more irreps, the hyperparameter-task of choosing representation in each layer is also more difficult. We believe that a promising direction for future research on scaling up efficient equivariant networks is to work with larger dihedral groups such as $D_8$—the symmetry group of the square pixel grid. $D_8$ has four irreps of dimension 1 and only one irrep of dimension $2 > 1$. Further, dihedral groups have the conceptual advantage that all of their irreps are of real type, so that the dimension of $\mathrm{Hom}(\varrho, \varrho)$ is 1 according to Schur's lemma.

## B. Experimental Setting

We train on images of size $224 \times 224$ throughout. Hyperparameters are listed in Table 2. While ConvNeXt uses exponential moving accuracy by default, we find for the equivariant nets that the non-averaged model gives better accuracy and therefore report that accuracy.

**Software versioning**. Our experiments build upon PyTorch (Paszke et al., 2019) and the timm (Wightman, 2019) library. We enable mixed-precision training using the deprecated NVIDIA library Apex, this is to mirror the training recipes of the benchmarks as closely as possible. To enable PyTorch's compiler, we use a modern version ($\geq 2.0$). Specifically, we use PyTorch 2.5.1 with CUDA 11.8.

**Hardware**. All experiments were run on NVIDIA A100-40GB. The per GPU batch size ranged between 64 (for larger models) to 256 (for smaller models). The biggest model requires training on 32 A100 GPUs for c. 2 days. The baselines were trained on V100 GPUs by the respective authors.

**Model sizes**. The model sizes, referred to in the paper as "Tiny" (T), "Small" (S), "Base" (B), "Large" (L), and "Huge" (H) are specified in Table 3.

**Accelerating throughput**. We use three tools to improve the throughput of the networks, both the baselines and the equivariant implementations.

1. Flash Attention (Dao, 2023), used by the ViTs.

2. Mixed-precision training using NVIDIA's Apex library, used by all models except the ResMLPs.

3. PyTorch's compiler and high precision matmul.

**Calculating throughput**. As is specified in Table 1, the throughput and peak memory are measured on a single A100-40GB GPU with batch size fixed to 64 and compiled networks running mixed precision forward passes with no gradients. Moreover, we utilize 10 warm-up iterations and then average over 100 runs. To measure peak memory we make use of PyTorch's built in device memory allocation monitor.

**Counting FLOPs**. For counting the number of FLOPs, we use `fvcore.nn.FlopCountAnalysis` (fvcore). We further make sure to add support for the operations added by the flopping-equivariant implementation as the default counter in `fvcore` does not count elementwise additions for instance. FLOPs are normalized with respect to the batch size.

## C. ResMLP Linear Layers Over Patches

To make the linear layers over the patch dimension flopping-equivariant, we need to keep track of what happens with the patches as the image is flopped. If we denote by $x_1$ the $N \times N \times d/2$ tensor of invariant patch embeddings, then $x_1[:, k-1]$ changes place with $x_1[:, -k]$ when the image is flopped (where we use NumPy-style tensor indexing). Thus,

$$x_{1,1}[:, k-1] := x_1[:, k-1] + x_1[:, -k]$$

is actually an invariant quantity while

$$x_{1,-1}[:, k-1] := x_1[:, k-1] - x_1[:, -k]$$

is $(-1)$-equivariant. Similarly, from the $(-1)$-equivariant output from PatchEmbed, $x_{-1}$, we can construct the $(-1)$-equivariant

$$x_{-1,1}[:, k-1] := x_{-1}[:, k-1] + x_{-1}[:, -k]$$

and the invariant

$$x_{-1,-1}[:, k-1] := x_{-1}[:, k-1] - x_{-1}[:, -k].$$

By keeping track of the four tensors $x_{\pm 1, \pm 1}$, each of shape $(N \times N/2) \times d/2$, we can make both the linear layers over

Table 2: Training recipes for different model architectures. We try to, as closely as possible, replicate the training recipe of the baselines.

| Model → | DeiT III (Touvron et al., 2022) | | | | ResMLP (Touvron et al., 2023) | | | | ConvNeXt (Liu et al., 2022) | | |
|---|---|---|---|---|---|---|---|---|---|---|---|
| | ViT-S | ViT-B | ViT-L | ViT-H | ResMLP-T | ResMLP-S | ResMLP-B | ResMLP-L | ConvNeXt-S (*iso.*) | ConvNeXt-B (*iso.*) | ConvNeXt-L (*iso.*) |
| Batch size | 2048 | 2048 | 2048 | 2048 | 2048 | 2048 | 2048 | 2048 | 4096 | 4096 | 4096 |
| Optimizer | LAMB | LAMB | LAMB | LAMB | LAMB | LAMB | LAMB | LAMB | AdamW | AdamW | AdamW |
| LR | $3.10^{-3}$ | $3.10^{-3}$ | $3.10^{-3}$ | $3.10^{-3}$ | $5.10^{-3}$ | $5.10^{-3}$ | $5.10^{-4}$ | $5.10^{-4}$ | $4.10^{-3}$ | $4.10^{-3}$ | $4.10^{-3}$ |
| LR decay | cosine | cosine | cosine | cosine | cosine | cosine | cosine | cosine | cosine | cosine | cosine |
| Weight decay | 0.02 | 0.02 | 0.02 | 0.02 | 0.2 | 0.2 | 0.2 | 0.2 | 0.05 | 0.05 | 0.05 |
| Training epochs | 400 | 400 | 400 | 400 | 400 | 400 | 400 | 400 | 300 | 300 | 300 |
| Warmup epochs | 5 | 5 | 5 | 5 | 5 | 5 | 5 | 5 | 50 | 50 | 50 |
| Label smoothing $\varepsilon$ | ✗ | ✗ | ✗ | ✗ | 0.1 | 0.1 | 0.1 | 0.1 | 0.1 | 0.1 | 0.1 |
| Dropout | ✗ | ✗ | ✗ | ✗ | ✗ | ✗ | ✗ | ✗ | ✗ | ✗ | ✗ |
| Stoch. Depth | 0.0 | 0.1 | 0.4 | 0.5 | 0.05 | 0.2 | 0.2 | 0.2 | 0.4 | 0.5 | 0.5 |
| Repeated Aug | ✓ | ✓ | ✓ | ✓ | ✓ | ✓ | ✓ | ✓ | ✗ | ✗ | ✗ |
| Gradient Clip. | 1.0 | 1.0 | 1.0 | 1.0 | ✗ | ✗ | ✗ | ✗ | ✗ | ✗ | ✗ |
| Mixed Precision | ✓ | ✓ | ✓ | ✓ | ✗ | ✗ | ✗ | ✗ | ✓ | ✓ | ✓ |
| Rand Augment | ✗ | ✗ | ✗ | ✗ | ✓ | ✓ | ✓ | ✓ | ✓ | ✓ | ✓ |
| 3 Augment | ✓ | ✓ | ✓ | ✓ | ✗ | ✗ | ✗ | ✗ | ✗ | ✗ | ✗ |
| Mixup alpha | 0.8 | 0.8 | 0.8 | 0.8 | 0.8 | 0.8 | 0.8 | 0.8 | 0.8 | 0.8 | 0.8 |
| Cutmix alpha | 1.0 | 1.0 | 1.0 | 1.0 | 1.0 | 1.0 | 1.0 | 1.0 | 1.0 | 1.0 | 1.0 |
| Erasing prob. | ✗ | ✗ | ✗ | ✗ | 0.25 | 0.25 | 0.25 | 0.25 | 0.25 | 0.25 | 0.25 |
| ColorJitter | 0.3 | 0.3 | 0.3 | 0.3 | 0.3 | 0.3 | 0.3 | 0.3 | 0.4 | 0.4 | 0.4 |
| Test crop ratio | 1.0 | 1.0 | 1.0 | 1.0 | 1.0 | 1.0 | 1.0 | 1.0 | 1.0 | 1.0 | 1.0 |
| Loss | BCE | BCE | BCE | BCE | CE | CE | CE | CE | CE | CE | CE |

Table 3: Specification of model notation in terms of depth and width (embedding dimension).

| Architecture | Depth | Width |
|---|---|---|
| **MLP-Mixers (ResMLP (Touvron et al., 2023))** | | |
| ResMLP-L | 24 | 1280 |
| ResMLP-B | 24 | 768 |
| ResMLP-S | 24 | 384 |
| ResMLP-T | 12 | 384 |
| **Vision Transformers (DeiT III (Touvron et al., 2022))** | | |
| ViT-H | 32 | 1280 |
| ViT-L | 24 | 1024 |
| ViT-B | 12 | 768 |
| ViT-S | 12 | 384 |
| **Convolutional Networks (ConvNeXt (Liu et al., 2022))** | | |
| ConvNeXt-L (*iso.*) | 36 | 1024 |
| ConvNeXt-B (*iso.*) | 18 | 768 |
| ConvNeXt-S (*iso.*) | 18 | 384 |

the patch dimension $(N \times N)$ and the channel dimension $d$ efficient through the decomposition (2).

This calculation is an example of a Clebsch-Gordan product or tensor product of representations, which has been used in some prior designs of equivariant networks (Kondor et al., 2018).

