# OpenReview forum: "Flopping for FLOPs: Leveraging Equivariance for Computational Efficiency"
_ICML.cc/2025/Conference — ICML 2025 spotlightposter_

### Official Review · Reviewer_RXrY · 2025-03-04

**Overall Recommendation:** 4

**Summary:**

The paper introduces three new equivariant neural networks (adapted from three non-equivariant models: ResMLP, ViT and ConvNeXt) for the purpose of showing that the equivariant models maintain a comparable number of floating-point operations (FLOPs) per parameter versus their non-equivariant counterparts. The main idea behind the authors' work is that they replace linear layers with block diagonal equivariant linear layers using irreps of the group C_2 (the flopping group), which ultimately parameterise feature spaces in terms of mirror-symmetric and mirror-antisymmetric features. They describe how to obtain invariant and flopping equivariant features using a patch embedding layer, how to implement the non-linear equivariant layers, and how to adapt self-attention to obtain equivariance. The authors evaluate their models on the ImageNet-1K dataset and compare them based on three measures: number of parameters, FLOPs and throughput.

**Claims And Evidence:**

The claims appear to be well-founded, with well-explained supporting evidence. However, as stated below, I question the extent of the contribution made, given that the analysis that appears only considers equivariance to the group C_2.

**Essential References Not Discussed:**

No, I didn't pick up any important ones.

**Experimental Designs Or Analyses:**

I read about the design of all of the experiments, and found them to be sound, with fair comparisons being made. In particular, I liked the explanation behind the authors' decision to consider the three measures that they did, and thought that their argument was well-considered.

**Methods And Evaluation Criteria:**

Given the question that the authors posed throughout, I thought that the proposed methods and benchmarks made sense for the problem at hand.

**Other Comments Or Suggestions:**

I have a big gripe about the sentence given in lines 46-48, 2nd column: "In a nutshell, our case is that equivariant neural networks are simple models that scale well." For me, this claim as stated is far beyond what is presented in this paper: indeed, there are many more equivariant neural networks that exist for far many more groups beyond C_2 in the literature. I would strongly recommend that the authors adjust this sentence in a camera-ready version of this paper should it be accepted.

I think that the Patch embedding layer could be described better - it is not entirely clear from Figure 3 how this works. Could the authors provide a clear example of this procedure, perhaps even in the appendix?

This is a small point but I would personally move Figures 1 and 2 to the second page - it looks a bit messy on the first page.

**Other Strengths And Weaknesses:**

Strengths: I thought that the paper was well structured and easy to read, and I commend the authors for achieving this. The question of achieving scalable equivariant models that do not dramatically increase computational costs is an important one that needs further study in the equivariant community. The authors clearly demonstrate that the equivariant models that they have considered achieve a comparable number of floating-point operations per parameter to the baseline ordinary neural networks. I commend the authors for providing a clear example where equivariant models can be scaled whilst maintaining their usefulness.

Weaknesses: Despite this, I question the depth of the originality behind these results, given that the paper targets only one type of equivariance whose symmetries, beyond perhaps the data studied (image data), are maybe not so important - this is my biggest concern behind this paper. However, I am open to the authors addressing this point of mine in their rebuttal if they can provide me with convincing evidence to the contrary. Complementary to my previous point, I think that the commentary given in lines 676-689, 1st column, needs to be given instead in the main paper and discussed in much more detail. In particular, I think that the authors need to discuss whether they think that the conclusion that they give in this paper (about the comparability of FLOPs per parameter) would extend to groups with the more "difficult" irreps. If what they have presented here is only a special case, then that would, in my view, reduce the significance of the present contribution. Finally, the way that the equivariant linear layers are constructed use an equal number of invariant and flopping equivariant features: as the authors themselves say in lines 181-184, 2nd column, it is not obvious that this is optimal. What happens if they are not equal? Does the entire analysis/construction of the models break down? It seems to me that it might do, but again, I am open to the authors clarifying this for me.

**Questions For Authors:**

I have provided the questions that I would need answering to be able to adjust my score in the weaknesses section. To repeat them here, they focus on:

1) Could the authors provide other examples where their C_2 equivariant construction might be useful beyond the image data they have experimented with?

2) Do the authors think that the FLOPs results that they have presented here for C_2 equivariant models extend to groups with more difficult irreps? I am concerned, given the way that the authors have presented their work, that the results only work for this group (and this situation) as a special case; however, as stated above, I am open to being told otherwise.

3) What happens to the analysis provided in the paper if the number of invariant and flopping equivariant features is not equal? Can their construction still be used, or not? If not, how likely would one use an equal number of these features?

EDIT: Following the authors' rebuttal, I have decided to raise my score from 3 to 4.

**Relation To Broader Scientific Literature:**

I found that the authors set their paper well in the context of the broader scientific literature, in particular, with a well-framed related work section. They set their work in the context of geometric deep learning (Bronstein et al. 2021), considered the numerous works behind steerable equivariant ConvNets, and highlighted the differences behind their ViT networks and those used by Kundu and Kondor (2024). I also enjoyed the discussion behind equivariant networks for other input than images, although I note that the group considered in these related works is different from the one under consideration in this paper (there is, however, no issue with this).

**Theoretical Claims:**

I checked the correctness of the derivation of the linear layers in Section 3.1, and deemed it to be correct. I also checked the equivariant version of the pointwise activation in (3), and confirmed that it is correct.

---

> ### Author Rebuttal · Authors · 2025-03-31
>
> We thank the reviewer for the thorough review. We address their concerns here, which will sharpen the paper.
>
> > [...] the version of Schur's Lemma (A.2) provided by the authors was a bit overkill [...]
>
> We are unaware of simpler versions of Schur’s Lemma for real irreps of finite groups than the one given. If the reviewer has a simpler form in mind, we welcome any suggestions.
>
> > Q1: Could the authors provide other examples where their C_2 equivariant construction might be useful [...]?
>
> Since $C_2$ is the simplest possible symmetry group, it appears in many situations. Examples include
>
> - 3D tasks: Many natural and man-made objects (and parts of objects) are equally likely to appear mirrored. Thus, tasks involving these are often mirror equivariant/invariant. This includes data such as meshes, point clouds, or 3D images.
> - Many games have mirror symmetry. E.g., for training neural networks to play Go, it is common to apply reflection augmentation.
> - Mirror equivariance/invariance is also common in 1D data. E.g., measurements such as soil samples along a path should be classified the same regardless of the path's direction.
>
> We believe that experiments on images are sufficient to demonstrate the idea and that the new architectures are a valuable contribution to the literature on vision models.
>
> > Q2 [and earlier comments]: Do the authors think that the FLOPs results that they have presented here for C_2 equivariant models extend to groups with more difficult irreps?
>
> The paper makes two different points.
>
> The first point is that the FLOPs-per-parameter ratio is exactly the same for non-equivariant and equivariant linear layers with the same input and output feature dimensions. This only holds when the irreps are one-dimensional, as mentioned in Appendix A. We will go through the main paper and make it clearer there. For instance, as pointed out by Reviewer s6u9, there is a formulation in Section 5 where “equivariant networks can be designed to have the same number of FLOPs…” will be changed to “flopping equivariant networks…”. We do not think we have over-claimed the contribution since the paper's title already clarifies that we mainly consider the flopping symmetry in this work.
>
> The second point is that hard-coding equivariance results in computational savings, due to block-diagonalization of the linear layers. This argument generalizes straightforwardly to any other groups due to Schur’s lemma. In particular, for groups with more than two irreps, layers mapping between feature spaces transforming according to (multiples of the isotypical decomposition of) the regular representation will decompose into more blocks, resulting in larger computational savings. To rephrase slightly, the FLOPs-per-feature is much reduced in linear layers with hard-coded equivariance, which generalises to all finite groups.
>
> We think the Reviewer’s suggestion to include a discussion around the generalisation to other groups in the main paper is a good idea, and we will do so.
>
> > Q3: What happens to the analysis provided in the paper if the number of invariant and flopping equivariant features is not equal?
>
> The block decomposition still works if we do not have an equal amount of invariant and (-1)-equivariant features. The 0-blocks in equation (2) will be smaller and so the computational savings per feature will be smaller. The number of FLOPs per parameter will still be the same.
>
> We have included an extra experiment in the rebuttal to Reviewer s6u9 (Q1) to test whether having more invariant features is helpful for the classification task.
>
> Having an equal amount corresponds to having features that transform according to the regular representation of $C_2$, but in a diagonal basis (the isotypical decomposition). Weiler & Cesa (2019) found that the regular representation is generally a good choice.
>
> > I have a big gripe about [...] "In a nutshell, our case is that equivariant neural networks are simple models that scale well."
>
> We thank the reviewer for pointing this out. Indeed, the intended meaning is not that all equivariant neural networks are simple and scale well. We will revise the sentence: "In a nutshell, our case is that equivariant neural networks *can be* simple models that scale well.”
>
> > I think that the Patch embedding layer could be described better [...]
>
> Consider the case of 1D-signals: We have a signal [a, b, c, d] and its flopped version is [d, c, b, a]. If we use a symmetric filter [v, v] with stride 2, we get [v(a + b), v(c + d)] and [v(d + c), v(b + a)] respectively. Thus, when the input is flopped, the feature map is flopped too. If we use an antisymmetric filter [w, -w], we get [w(a - b), w(c - d)] and [w(d - c), w(b - a)] respectively. Thus, when the input is flopped, the feature map is flopped and changes sign.
>
> In the PatchEmbed-layer we use equally many symmetric and antisymmetric filters to obtain invariant and (-1)-equivariant features. As mentioned above, this is a common choice.

---

> > ### Comment · Reviewer_RXrY · 2025-04-02
> >
> > I thank the authors for their detailed rebuttal. I have read all of the reviews and corresponding rebuttals. I retract my comment about their statement of Schur's Lemma - I must have misread it in my original reading - and am happy with what they have presented in their paper. I would particularly like the authors to include the additional examples in the main text - this will help demonstrate the potential impact of their work. I am glad the authors will include a discussion around the generalisation to other groups in the main paper - again, for potential impact, I think that this is important.
> >
> > The authors wrote: "We do not think we have over-claimed the contribution since the paper's title already clarifies that we mainly consider the flopping symmetry in this work." I agree that although in principle they have not overclaimed the contribution, people easily forget the title, and so I think it is important, especially in making any type of claim, that the authors are precise in what they are saying. However, I do note that they are going to edit the "gripe" I picked up and the one that Reviewer s6u9 picked up, which I welcome.
> >
> > In summary, I have decided that, so long as the commitments to making the proposed changes appear in a revised version of this paper, I am happy to raise my score from 3 to 4.

---

> > > ### Author Response · Authors · 2025-04-04
> > >
> > > We thank the reviewer for their response and confirm that we commit to making the proposed changes in a camera-ready version of the paper if it is accepted to the conference.

---

### Official Review · Reviewer_debe · 2025-03-14

**Overall Recommendation:** 5

**Summary:**

The paper suggests a clever implementation of flop-equivariant linear layers - together with some strategies to adapt other layers too - which allows to achieve flop-equivariance in most popular vision architecture while halving the computation cost with respect to their non-equivariant counterparts.
The paper is very well written and motivated and includes extensive empirical analysis.

**Claims And Evidence:**

All claims seem well supported by empirical evidence and theoretical arguments.
Moreover, all the assumptions are backed up by extensively citing many previous works which provided relevant insights.
Overall, this makes the story of the paper very convincing.

**Essential References Not Discussed:**

N/A

**Experimental Designs Or Analyses:**

See above

**Methods And Evaluation Criteria:**

See above

**Other Comments Or Suggestions:**

The block diagonal decomposition of the linear layers in Sec 3.1 doesn't directly work for convolution layers, unless one only considers flop-invariant filters since anti-symmetric filters can map between the two types of features.
This is briefly discussed later in Sec 4.3 but I think the authors could expand this point earlier in the paper for improved clarity.

**Other Strengths And Weaknesses:**

The paper is really well motivated, with a very convincing story and introduction.
In particular, I really enjoyed reading Sec 3.5, which includes some very elegant arguments and provides in a clear and concise way many practical insights about the design of equivariant architectures.
I would even suggest the authors to mention some of these points in the introduction as part of the contribution.

Sec 3.1 also manages to give an intuitive yet precise description of equivariant layers and the potential computational gain, making the main ideas in the paper accessible for a wider audience.

I do not see any significant weaknesses in this work.
Of course, the empirical evaluation could be made stronger by scaling further and performing hyper-parameter tuning of the equivariant models as already argued by the authors, but I agree this is beyond the scope of this paper.
I am looking forward to see the community picking up on these ideas and further extend the empirical validation.

**Questions For Authors:**

N/A

**Relation To Broader Scientific Literature:**

The manuscript properly places itself in the literature, extensively citing the long line of works which provided the insights used to motivate this work.

**Theoretical Claims:**

See above

---

> ### Author Rebuttal · Authors · 2025-03-31
>
> We thank the reviewer for the review and for highlighting Section 3.5. We are pleased to read that the reviewer finds the paper valuable.
>
> > The block diagonal decomposition of the linear layers in Sec 3.1 doesn't directly work for convolution layers, unless one only considers flop-invariant filters since anti-symmetric filters can map between the two types of features. This is briefly discussed later in Sec 4.3 but I think the authors could expand this point earlier in the paper for improved clarity.
>
> We agree with this comment and will modify Section 3.1. We thank the reviewer for the suggestion.
>
> Finally, we have included one extra experiment in the rebuttal to reviewer s6u9 (Q1), which could interest all reviewers.

---

### Official Review · Reviewer_s6u9 · 2025-03-17

**Overall Recommendation:** 3

**Summary:**

In this paper, the authors present a flopping equivariant variant of known vision models like ConvNext, ViTs, and ResMLP. The key idea is to parameterize feature space in terms of mirror symmetry and mirror anti-symmetry features. This approach reduces FLOPs and wall-clock time to give an efficient scalable and efficient symmetry architecture. The empirical experiments are done on the Imagenet 1K dataset Flops, throughput, and number of parameters are reported for three different architectures.

## post rebuttal
I will keep my score as the authors gave a satisfactory rebuttal.

**Claims And Evidence:**

The claim in section 5: *equivariant networks can be designed to have the same number of FLOPs...* seems to lack evidence, as the paper only shows this for a flop equivariance, and this might not be true for higher dihedral groups.

**Essential References Not Discussed:**

Yes, most of the related works are covered in the paper. Although a few important works regarding are missing.


1. Structuring Representation Geometry with Rotationally Equivariant Contrastive Learning, Gupta et al.
2. On the Utility of Equivariance and Symmetry Breaking in Deep Learning Architectures on Point Clouds, Vadgama et al.
3. On genuine invariance learning without weight-tying, Moskalev et al.
4. Equivariant Adaptation of Large Pretrained Models, Mondal et al.
5. Equivariance with learned canonicalization functions, Kaba et al

**Experimental Designs Or Analyses:**

Yes, all of the experiments were presented.

The experimental design seems reasonable, although there are a few questions regarding the broader use/ implications of the choice made in the paper. The relation to contrastive learning (with equivariance) as well as learned canonicalization especially post hoc way in models are some ways interesting comparisons to the proposed approach as they are more commonly used for images. [1,2] show that augmentation helps with performance even with equivariant models, and given the task of image classification, augmentations are not used in the pipeline.

The classification task which is an invariant task with respect to transformations is considered, and tasks like image editing which is equivariant to transformations are not considered without any justification.

1. Structuring Representation Geometry with Rotationally Equivariant Contrastive Learning, Gupta et al.
2. On the Utility of Equivariance and Symmetry Breaking in Deep Learning Architectures on Point Clouds, Vadgama et al.

**Methods And Evaluation Criteria:**

Yes, ImageNet 1K makes sense as well and the evaluation criteria are reasonable for the problem.

**Other Comments Or Suggestions:**

- In section 3.5, Limitations: It is unclear what the limitations described in the first paragraph.

- In section 3.5, important works on relaxing equivariance have not been mentioned. A lot of symmetry-breaking and relaxed equivariance works have been done for the domain of point clouds (molecules etc). See the Essential References section.

**Other Strengths And Weaknesses:**

## Strengths
- This paper presents a way to scale equivariant models by considering flop equivariant neural networks.
- The proposed approach is used for three different architectural choices: ViT, ConvNext, and ResMLPs which provides a good understanding of image classification using these architectures with their equivariant and hybrid counterparts.



## Weaknesses
- The paper does not include any suggestions for effective improvement in the pipeline like augmentation or generalizability to other group.
- The low performance of smaller equivariant models is not justified.

**Questions For Authors:**

1. The equal split of invariant and (-1)-equivariant features are used in the paper. How would the framework’s performance change if this ratio were adjusted based on task-specific learned symmetries and is there a way to learn the optimal division for?

2. Is it possible to generalize this approach beyond D2 (flopping)? How does that affect the performance with similar FLOPs?

3. Methods like learned canonicalization and its adaptation to large models improve performance without huge computational costs [1, 2]. How does the proposed approach compare to these techniques?

4. Does data augmentation like rotations or flips (commonly used in contrastive learning) help with performance in the proposed approach?

5. With libraries like Cu-equivariance [3] which provides improved irreps computation with cuda implementation, how does this affect the impact of the proposed work?

[1] Equivariant Adaptation of Large Pretrained Models, Mondal et al.

[2] Equivariance with learned canonicalization functions, Kaba et al

[3] [cu-equivariance](https://github.com/NVIDIA/cuEquivariance)

and see weaknesses

**Relation To Broader Scientific Literature:**

The key contribution of the paper is to scale equivariant models using mirror symmetry and mirror anti-symmetry is interesting and is novel, although the idea of using invariants to design equivariant networks is not uncommon.
Important links to existing works in a slightly different domain (point clouds, contrastive learning approach) are missing and also effective use for image classification tasks seems superfluous as ViT with post hoc canonilication or conservative learning transformation pipeline can achieve similar results.

The important takeaway from the paper is that equivariant models can be scaled in the proposed manner, and there could be more use cases beyond the ones mentioned in the paper.

**Theoretical Claims:**

Yes, they seem correct.

---

> ### Author Rebuttal · Authors · 2025-03-31
>
> We thank the reviewer for the helpful review and relevant questions.
>
> > The claim in section 5: equivariant networks can be designed to have the same number of FLOPs... seems to lack evidence [...].
>
> We thank the reviewer for the pointer and will refine it to “flopping equivariant networks”.
>
> > Essential References Not Discussed
>
> The suggested references are relevant and will be included. Note that our idea for speeding up equivariant networks has not been applied in any of these papers.
>
> > In section 3.5, Limitations: It is unclear what the limitations described in the first paragraph.
>
> One argument against hard-coding equivariance is that it limits what features can be learned. We argue that this limitation is weak in the first paragraph. We will make it clearer.
>
> > Q1. [...] How would the framework’s performance change if this ratio were adjusted based on task-specific learned symmetries and is there a way to learn the optimal division for?
>
> To answer this question, we have implemented a ViT that has an equal split of invariant and (-1)-equivariant features for the first half of the network and only invariant features for the second half of the network (the (-1)-equivariant features from the first half of the network are invariarized by taking their absolute value, incidentally this is a “complete invariant”). For layers with only invariant features, the block-decomposition in equation (2) turns into a dense matrix $W_{1,1}$ since $W_{-1,-1}$ doesn’t exist. So, there are no FLOP savings compared to ordinary layers, and the FLOP-per-parameter ratio is also the same.
>
> The accuracies on ImageNet are as follows:
>
> ViT-S: no convergence, ViT-B: 82.5%, ViT-L: 84.1%, ViT-H: 84.7%.
>
> Comparing these with the results in Table 1, we see that they consistently outperform the equivariant ViT with equal amounts of invariant and (-1)-equivariant features. This aligns with the prior work as discussed in Section 3.5. In terms of total FLOPs and parameters the new networks are between the previous equivariant and the baseline networks, for ViT-H the values are almost exactly the same as the previously considered hybrid network.
>
> We are not aware of a method to guarantee learning the optimal division of irreps, but as outlined in Section 3.5, we regard this as promising and interesting future work.
>
> > Q2 [and earlier comments]. Is it possible to generalize this approach beyond D2 (flopping)? How does that affect the performance with similar FLOPs?
>
> Yes, this is possible. We discuss this extension in Appendix A. The FLOPs-per-parameter ratio depends on the group and it is always possible to achieve computational savings. Please refer to the rebuttal to Reviewer RXrY (Q2) for a detailed discussion.
>
> > Q3. Methods like learned canonicalization and its adaptation to large models improve performance without huge computational costs [1, 2]. How does the proposed approach compare to these techniques?
>
> Canonicalization would mean training an extra network that maps an image to a canonical orientation and then feeding that image into a pretrained network. This is useful when a pretrained network exists which is not invariant. A common example would be a network trained on upright images not generalising to rotated images. If the data that the pretrained network was trained on already contains the symmetry (e.g. upright images are equally common in flopped format) then canonicalization can not improve performance because the pretrained network is not better on a particular orientation of the images. Also, highly symmetric images can pose a problem for canonicalization-based methods (see e.g. Appendix A of the mentioned [2]). Finally, canonicalization methods incur an extra overhead of running the canonicalization network, while we instead obtain networks that run faster than the baseline networks.
>
> > Q4. Does data augmentation like rotations or flips (commonly used in contrastive learning) help with performance in the proposed approach?
>
> Yes, we use heavy data augmentation like the baseline methods, including Mixup, Cutmix, etc. Refer to Table 2 in the appendix.
>
> > Q5. With libraries like Cu-equivariance [3] which provides improved irreps computation with cuda implementation, how does this affect the impact of the proposed work?
>
> Libraries with faster equivariant layers further strengthen the broad case for equivariant networks made in Section 3. Cu-equivariance is specialised for processing 3D graphs with rotation ambiguity, such as molecular data, and thus it is orthogonal to our implementations for images.

---

### Official Review · Reviewer_3YAc · 2025-03-18

**Overall Recommendation:** 3

**Summary:**

The paper's main goal is to develop new equivariant vision models that scale effectively. It presents equivariant networks that preserve mirror symmetry while keeping FLOPs comparable to non-equivariant models. The focus is on simple image symmetries for more efficient computation. The proposed network divides all feature maps into two types: flopping invariant features and flopping equivariant features. The latter undergo a sign flip when mirroring occurs. This split architecture enables smaller matrix multiplication while maintaining equivariance, cutting the number of FLOPs in half. The paper discusses a patch embedding layer along with considerations for non-linearities and attention. It examines three successful vision architectures: ResMLP (Touvron et al., 2023), ViT (Dosovitskiy et al., 2021), and ConvNeXt (Liu et al., 2022). Each architecture is explained simply, with adaptations proposed using the flopping equivariant architecture. The experiments compare: 1) model size versus FLOPs, 2) network throughput, and 3) memory usage. Results for all architectures are presented on ImageNet, showing that the proposed models scale well compared to state-of-the-art non-equivariant vision models.

**Claims And Evidence:**

The main motivation of the paper is to encourage work on scaling equivariant neural networks. The main claims are 1) flopping equivariant networks can achieve accuracy comparable to the state of the art vision networks. 2) the number of FLOPs vs model size tradeoff is comparable. 3) The memory footprint is reduced. The results in figure 5 and table 1 do accurately support the main claims of the paper.

**Essential References Not Discussed:**

I think references are discussed adequately. However, I would like to suggest the authors also take a look at the following paper: "Symmetry-Based Structured Matrices for Efficient Approximately Equivariant Networks" by Samudre et al. This paper is about exploring a similar theme, even though the paper is mainly theoretical, and about approximate equivariance. Note that the experiments seem to have a typo (FLOPS instead of FLOPs).

**Experimental Designs Or Analyses:**

The experimental design and the choice of metric is sensible. The appendix describes experimental recipes in more detail.

**Methods And Evaluation Criteria:**

Yes, the evaluation criteria (comparison in terms of accuracy, throughput, memory, FLOPs, model size) are sensible. The three baseline architectures are well-optimized in the literature on ImageNet, so the comparison actually puts the proposed model at a slight disadvantage.

**Other Comments Or Suggestions:**

N/A

**Other Strengths And Weaknesses:**

As such, the ideas used in the paper are technically straightforward. However, its main goal is to encourage designing equivariant models that have favourable scaling properties. In that goal, I believe the paper does a decent job.

**Questions For Authors:**

Could the authors expand on why implementing the Winograd scheme might be involved on GPU?

**Relation To Broader Scientific Literature:**

Exploring the scaling behaviour of equivariant models is only a recent area of interest. This paper considers specific vision models and then proposes equivariant counterparts that scale well. I think the key contribution is well-motivated and of broader interest.

**Theoretical Claims:**

Not applicable.

---

> ### Author Rebuttal · Authors · 2025-03-31
>
> We thank the reviewer for the clear review, which accurately summarises our main contributions.
>
> > I think references are discussed adequately. However, I would like to suggest the authors also take a look at the following paper: "Symmetry-Based Structured Matrices for Efficient Approximately Equivariant Networks" by Samudre et al. This paper is about exploring a similar theme, even though the paper is mainly theoretical, and about approximate equivariance. Note that the experiments seem to have a typo (FLOPS instead of FLOPs).
>
> This is a good suggestion for an additional reference, which we will add. Combining the structured matrix approach with the irrep-decomposition approach could be interesting for future work.
>
> > As such, the ideas used in the paper are technically straightforward. However, its main goal is to encourage designing equivariant models that have favourable scaling properties. In that goal, I believe the paper does a decent job.
>
> We hope the technical straightforwardness can be regarded as a strength in this context.
>
> > Could the authors expand on why implementing the Winograd scheme might be involved on GPU?
>
> Implementing a naive version of the scheme is quite straightforward. What is challenging is making the implementation competitive with highly optimised GPU-kernels for ordinary convolutions, such as the ones in the widely used closed source cuDNN library. It requires careful handling of data shuffling within the GPU as well as proper utilization of hardware specifics such as tensor cores.
>
> Finally, we have included one extra experiment in the rebuttal to reviewer s6u9 (Q1), which may interest all reviewers.

---

> > ### Comment · Reviewer_3YAc · 2025-04-06
> >
> > Thank you for the concise clarifications, which I found useful. I would like to keep my recommendation of accept.

---

### Decision · Program_Chairs · 2025-05-01

**Decision:**

Accept (spotlight poster)

**Comment:**

This paper suggests to modify popular vision models by taking into account their equivariance to flopping (mirror symmetry). Importantly, the number of operations used within the network is reduced by two (in contrast with most equivariant models where typically compute time can be higher than non-equivariant models, though number of parameters is reduced). The reviewers were all happy with this paper and with the rebuttal. I ask the authors to remember to incorporate the improvements requested by the reviewers in the camera-ready version.